# A Modelling Framework Linking Resource-Based Stochastic Translation to the Optimal Design of Synthetic Constructs

**DOI:** 10.3390/biology10010037

**Published:** 2021-01-07

**Authors:** Peter Sarvari, Duncan Ingram, Guy-Bart Stan

**Affiliations:** 1Quantitative and Computational Biology, Dornsife College of Letters, Arts and Sciences, University of Southern California, Los Angeles, CA 90089, USA; sarvari@usc.edu; 2Imperial College Centre for Synthetic Biology, Imperial College London, London SW7 2BU, UK; d.ingram@imperial.ac.uk; 3Department of Bioengineering, Imperial College London, London SW7 2BU, UK

**Keywords:** synthetic biology, whole-cell model, translation, stochastic simulation, TASEP, construct design, burden, ribosomal queues, slow codon

## Abstract

**Simple Summary:**

In synthetic biology, it is commonplace to design and insert gene expression constructs into cells for the production of useful proteins. In order to maximise production yield, it is useful to predict the performance of these “engineered cells” in advance of conducting experiments. This is typically a complex task, which in recent years has motivated the use of “whole-cell models” (WCMs) that act as computational tools for predicting different aspects of cell growth. Many useful WCMs exist, however a common problem is their over-simplification of ribosome movement on mRNA transcripts during translation. WCMs typically don’t consider that, for constructs with inefficient (“slow”) codons, ribosomes can stall and form “traffic jams”, thereby becoming unavailable for translation of other proteins. To more accurately address these scenarios, we have built a computational framework that combines whole-cell modelling with a detailed account of ribosome movement on mRNA. We show how our framework can be used to link the modular design of a gene expression construct (via its promoter, ribosome binding site and codon composition) to protein yield during continuous cell culture, with a particular focus on how the optimal design can change over time in the presence or absence of “slow” codons.

**Abstract:**

The effect of gene expression burden on engineered cells has motivated the use of “whole-cell models” (WCMs) that use shared cellular resources to predict how unnatural gene expression affects cell growth. A common problem with many WCMs is their inability to capture translation in sufficient detail to consider the impact of ribosomal queue formation on mRNA transcripts. To address this, we have built a “stochastic cell calculator” (StoCellAtor) that combines a modified TASEP with a stochastic implementation of an existing WCM. We show how our framework can be used to link a synthetic construct’s modular design (promoter, ribosome binding site (RBS) and codon composition) to protein yield during continuous culture, with a particular focus on the effects of low-efficiency codons and their impact on ribosomal queues. Through our analysis, we recover design principles previously established in our work on burden-sensing strategies, namely that changing promoter strength is often a more efficient way to increase protein yield than RBS strength. Importantly, however, we show how these design implications can change depending on both the duration of protein expression, and on the presence of ribosomal queues.

## 1. Introduction

### 1.1. Whole-Cell Models in Synthetic Biology

It is routine in synthetic biology that theoretical predictions misalign with experimental results, particularly when unnatural genes are highly expressed [1,2]. This phenomenon is often attributed to “gene expression burden”, the over-consumption of shared cellular resources due to “unnatural” gene expression (also referred to as “heterologous” or “synthetic” expression), which can lead to reduced cell growth or even cell death. In turn, this causes yields of heterologous proteins to drop, motivating the study of the relationship between heterologous protein production yield and cellular growth.

An effective method that allows researchers to explore how burden affects cellular growth is via the construction of resource-limited “whole-cell models” (WCMs), which typically aim to describe key cellular processes (e.g., transcription, translation, metabolism, etc.) given finite cellular resources, such as energy, amino acids, polymerases and ribosomes [3,4,5,6,7,8]. Within such frameworks, the expression of synthetic gene constructs drains resources that are required for normal cell growth, effectively coupling heterologous gene expression to the growth rate. This notion is increasingly being considered in today’s synthetic biology construct designs [9], allowing genetic engineers and synthetic biologists to better understand how tweaking the components of gene constructs (e.g., promoters, ribosome binding sites (RBSs), terminators, etc.) can reduce burden, and thereby improve cell growth and protein yield. Precisely how cell production is affected, however, is difficult to predict, and so, WCMs are often used to explore how various genetic construct designs affect the performance of engineered cells, e.g., in terms of the burden their expression imposes on their host cell and/or the dynamics of the protein production yields achievable by different designs.

### 1.2. Slow Codons and Ribosomal Queues

While they provide a useful foundation, existing WCMs fall short in capturing important biological phenomena, for example the movement of ribosomes during translation, which can often form “traffic jams” as they process an mRNA transcript. This queuing effect burdens the cell via wasteful sequestration of translational resources [10,11], and so limits the cell’s growth. One of the main features affecting ribosome movement along mRNAs is the codon composition of transcripts. Each codon type is associated with a different abundance of charged tRNA molecules, such that each is translated at a different rate [12]. This causes ribosomes to change speed and potentially form queues as they translate, an effect that is amplified when inefficient (“slow”) codons are present in transcripts. Therefore, the design efficiency of a construct in terms of its modular parts and codon composition has a potentially large impact on gene expression burden, cell growth and protein yield, and should ideally be considered in any whole-cell model that considers gene construct expression.

While the presence of slow codons on a transcript likely promotes queue formation and resource sequestration, their occurrence is not always bad for cell growth. For example, some organisms have been reported to use “ramp up” zones of slower codons at the 5’ end of their transcripts in order to stagger the elongation reactions and hence reduce the chances of costly upstream collisions and ribosomal queue formation [13,14,15]. A host of other evidence suggests that slowing ribosomes mid-translation can help with the fidelity of cotranslational folding [16,17], the process by which protein domains are organised into their correct tertiary structures while ribosomes are still translating [18].

Given these often-complex links between codon usage and protein yield, a host of computational tools is often used in order to optimise translation efficiency (many reviewed in [19], as well as others proposed in [20,21,22]). They typically rely on measures like the Codon Adaptation Index (a score that correlates codon usage bias with predicted heterologous gene expression efficiency) [23,24,25] and the Codon Context (a score denoting the optimisation of codon:anticodon pairing) [26,27,28], which while useful for obvious codon refinements, are typically unable to predict ribosomal queue formation. Additionally, genetic engineers are often limited in codon design by context-specific issues such as construct stability [29], meaning they do not have free-reign over their codon design. Given this, engineered transcripts in practice are rarely fully codon-optimised, and so, the use of slow codons does not usually benefit cell growth. Whether or not the effects are beneficial to the cell, it would be invaluable to be able to explore the whole-cell implications of slow synthetic codons in ribosomal stalling and queue formation.

### 1.3. Biophysical Models of Translation

While the aforementioned computational tools are able to correlate codon composition with basic estimates of protein yield efficiency, more detailed models of translation are required to understand the effects of ribosomal queues in the context of a growing cell. Inspiration can be taken from existing biophysical models of mRNA-ribosome interactions (many of which were reviewed in [30]), where known parameters and molecular interactions are used to build a realistic account of translation, without the need for extensive analysis of biological data [31,32,33]. This is in contrast to machine learning approaches, which have seen extensive use in practically predicting translation outputs from large sets of data [34,35,36], but nevertheless typically lack the ability to provide causal explanations for how each factor contributes to the output.

A broad range of biophysical translation models have been built in recent years that differ in their simulation method, complexity and use-case. The simplest of these rely on the tRNA Adaptation Index (tAI) [24,37,38], which assigns an efficiency to each codon principally based on (i) tRNA abundances and (ii) the thermodynamics of codon-anticodon pairing, and averages these across all codons of a gene. While methods that use the tAI have been shown to provide high performance in translation predictors [39,40,41], they lack the ability to describe how codon speeds vary across a transcript and, as such, cannot describe the effects of slow codons and their implications for ribosomal queues.

More detailed biophysical translation models not only consider individual codon efficiencies, but model the movement of ribosomes along mRNA transcripts, such that stalling and queue formation can be considered. Such models are typically described by the totally asymmetric simple exclusion process (TASEP), which considers mRNA transcripts as lattices upon which ribosomes move stochastically and unidirectionally using specific transition probabilities [42,43]. While the ideas for this theoretical framework were first envisioned decades ago, they have been progressively expanded and modified to provide detailed and sophisticated accounts of translation. The simplest TASEPs may model an individual “representative” mRNA transcript with an infinite supply of ribosomes and fixed efficiencies for each codon [44,45,46], while more intricate versions may consider different transcripts with unique codon profiles, dynamic pools of tRNAs and ribosomes or a broad range of experimental parameters about a cell’s physiology, among other aspects [15,47,48,49,50]. TASEPs have more recently been combined with organism-specific codon efficiencies and translation initiation rates to create online tools that expand upon those previously mentioned, such as in [21,22].

A limitation of all these models, however, is that their accounts of translation are disconnected from the cell’s other processes. This is in part due to the inherent complexity of running biophysical translation models alongside accurate descriptions of transcription, nutrient metabolism and cellular growth. Being able to achieve this would crucially enable us to study how ribosomal queues affect the balance between synthetic construct expression, gene expression burden and cellular growth, and in turn suggest how construct designs can be optimised in a range of useful experimental conditions.

### 1.4. A Combined WCM-TASEP Framework

In order to join codon-level translation with “whole-cell” dynamics, we hereafter present StoCellAtor, the “stochastic whole-cell calculator”. Our framework expands on our previous circuit-chassis models [1,51] by combining a simple TASEP framework with the general WCM of [4]. Such a WCM acts as a convenient platform to modify aspects of cellular growth by providing a resource-limited account of how transcription, translation and energy production affect cell growth. While other potentially suitable WCM frameworks exist [52], as far as we are aware, the modelling framework described in [4] is one of the simplest ones that offers a resource-limited description of cellular growth, and so is a suitable starting foundation with which to integrate a TASEP model of translation. When combining both parts of our framework, we made an additional modification to how the TASEP operates. Most TASEPs assign transition probabilities to all ribosomes on mRNA transcripts. To improve simulation efficiency, StoCellAtor only considers authorised transitions, i.e., forward transitions for non-queuing ribosomes.

In this paper we present our framework, conceptually illustrated in Figure 1, and show how it can be used to optimise protein yield by tweaking the design of a synthetic construct. We apply our model to simulate the heterologous protein production yield of an *E. coli* population when grown during continuous exponential growth and in a turbidostat that maintains constant cell density. In situations of minimal ribosome queuing, we find that increasing promoter strength or RBS strength has similar effects on boosting synthetic protein yield, whereas in the presence of ribosomal queues, increasing promoter strength rather than RBS strength appears to have a greater beneficial effect on heterologous protein yields. We then discuss how StoCellAtor can be expanded to analyse more complex transcript designs and, finally, how the impacts of construct genetic stability and mutation spread could be included.

## 2. Materials and Methods

### 2.1. Whole-Cell Model

Our model is structured and parameterised based on a stochastic version of the whole-cell framework by [4], with the core difference being the replacement of their one-step translation process with a multi-step TASEP. As in [4], we maintain the essential components of the model that capture how trade-offs between finite resources (energy supplies, free ribosomes and finite proteome) impact the cell’s growth rate ($G_{rate}$) during nutrient metabolism, transcription and translation. This model focuses on the effects of gene expression burden [1], as opposed to other varieties such as metabolic burden or toxicity. As such, it uses a simple account of nutrient metabolism, where the “quality” of the nutrients is determined by a single parameter, *n*. *n* is used to denote how many molecules of a general form of energy are produced from one molecule of intracellular nutrient (see the Appendix A for a detailed description of all parameters and variables).

In our description of the proteome, which is coarse-grained into different functional classes, we add specific parameters for the synthetic construct’s promoter strength (${\mathrm{prom}}_{H}$), RBS strength (${\mathrm{RBS}}_{H}$) and codon efficiencies. In order to combine the whole-cell components with a TASEP stochastic framework, we implement all processes using reaction-based rate equations, rather than deterministic differential equations. Each simulation is run until the heterologous protein variable reaches steady state, which we define as when it does not deviate by more than 1% from its mean during the last 10% of the simulation time. For a visualisation of each simulation’s convergence, see the Appendix A.

### 2.2. A Modified TASEP for Translation

Our TASEP implementation considers individual ribosome transitions along mRNA transcripts that belong to four classes: three of these are “endogenous” and therefore native to the cell (ribosomal (*R*), enzymatic (*E*), housekeeping (*Q*)), while one is unnaturally engineered into the cell (“heterologous” (*H*)). The lengths of transcripts are defined in terms of successive ribosomal footprints (R_f_), where 1 $R_{f}$ equates to 30 nucleotides [53], making each R_f_ account for 10 amino acids. As in [4], each transcript contains 30 successive footprints (900 nucleotides), except for *R* proteins, which contain 750 footprints (22,500 nucleotides), to reflect that ribosomes are multi-protein complexes requiring more resources to build [54,55]. While modelling mRNA degradation, “ribosome protection” is considered whereby transcripts cannot be degraded unless they are free from ribosomes. We focus our core results on a simple scenario that highlights the effects of ribosomal queues in order to clearly observe their impact. This illustrative scenario considers one slow codon with a relative efficiency of 0.5% at position 26 $R_{f}$ on a transcript of length 30 $R_{f}$. Other positions and efficiencies were also explored, and are reported in the Appendix A.

In each state transition, all bound ribosomes have a probability to transition to the next codon, with backwards transitions and detachments being neglected due to their rarity. The transition probability of each ribosome is proportional to the efficiency of the codon being translated, and so, by implementing codons with varying efficiencies, we can simulate the presence of “slow codons” and hence the formation of ribosomal queues. If a ribosome is directly behind another, its forward transition probability is recorded as zero such that it cannot be selected for a transition. This is a key difference with classical TASEPs, which would expend computational time first selecting a queuing ribosome and later finding it cannot move [56] (Figure 2a). Once a ribosome reaches the last codon of a transcript, one further elongation step releases it to create a protein molecule. Figure 2b shows how this translation framework is embedded in the wider whole-cell model, while Figure 2c displays a top-down perspective of all processes, highlighting the qualitative relationship between the cell’s native machinery, its heterologous protein production and its growth.

### 2.3. Model Use Cases

To apply our model to relevant experimental settings, we implement an analysis pipeline that uses steady-state simulation values to explore the impact of a construct’s design (promoter strength, RBS strength and codon composition) on the growth rate ($G_{rate}$) and heterologous protein production rate ($H_{rate}$) (Figure 3). We then use these values to calculate the protein yield that could theoretically be obtained over time in a growing cell population in two scenarios: uncapped exponential growth and growth within a turbidostat at steady state. The former provides insight into how dynamics evolve when there are no growth limitations, while the latter gives an insight into typical continuous culture settings where cell density is kept constant by adjusting the dilution rate. Depending on the experimental scenario, our analysis could be applied to other forms of continuous culture, for example a chemostat where the population’s growth rate is maintained constant by adjusting the nutrient concentration. However, we wanted to account for scenarios where the growth rate of a population may change mid-experiment, such as mutations occurring to the synthetic construct. In this case, a chemostat would change the nutrient concentration and in turn affect the cell density in order to reset the growth rate, while the turbidostat would simply adjust the dilution rate to keep the cell density constant.

### 2.4. Software

Our model was written in MATLAB 2018b, and the code was run on a computer equipped with 2 Intel® Xeon® E5-2670 2.60 GHz CPUs, 132 GB memory and the Ubuntu 14.04.5 LTS operating system. The scripts are available from GitHub at https://github.com/sarvarip/BacterialCellModel/tree/master/new_setting.

## 3. Results

### 3.1. Reproducing Growth Laws

Bacterial growth laws describe empirical relationships between a cell’s growth rate and another quantity of interest [57]. They are often used to check the validity of simulation results from whole-cell models. We first compare StoCellAtor’s “endogenous” output (without synthetic gene expression) with Monod’s and Schaechter’s laws to show that our model displays the typical cell function. We then compare our “heterologous” output with experimental trends reported by [3] to show its validity in capturing basic behaviours observed experimentally in the presence of synthetic gene expression.

Monod’s law describes a hyperbolic relationship between the concentration of the external nutrient and the growth rate [58], which we recover by varying the parameter for nutrient quality, *n*, which acts as a proxy for the external nutrient concentration (see Section 2.1). We run simulations with seven increasing values of *n* and record the steady-state $G_{rate}$, finding that it indeed saturates at higher values of *n* (Figure 4a), as in normal bacterial growth. For the second set of endogenous simulations, we compare the mRNA:protein mass ratio (see Appendix A) with the cell’s $G_{rate}$, a relationship that has been experimentally shown to be linear via “Schaechter’s law” [59]. We recover this trend by calculating the mass ratio at steady state for the different simulations considering various values of *n* and comparing these results with $G_{rate}$.

Higher expression levels of heterologous (*H*-class) proteins are known to lower a cell’s growth rate by reducing the amount of cellular resources available for the production of other proteins required for growth (e.g., ribosomal, enzymatic). Experimental results from [3] showed that this relationship is predominantly linear. For our heterologous simulations with uniform codon efficiency, we vary the cellular mass fraction of *H* ($H_{frac}$) by using nine different combinations of promoter and RBS strengths with n=100 (see Section 3.2). For each, we record steady-state protein quantities and $G_{rate}$, finding a strong linear relationship between them (Figure 4c).

### 3.2. Optimising Construct Design

#### 3.2.1. Relationships between Construct Design, Cell Growth and Heterologous Protein Yield

StoCellAtor can be used to explore the relationship between ribosomal queues, synthetic construct expression and cell growth. A key application from this is predicting the optimal design of synthetic constructs in terms of three elements: promoter strength (${\mathrm{prom}}_{H}$), RBS strength (${\mathrm{RBS}}_{H}$) and codon composition.

To gain insight into the impact of these parameters, we ran simulations for three values of both ${\mathrm{prom}}_{H}$ and ${\mathrm{RBS}}_{H}$ ($\frac{1}{3}$, 1 and 3), giving nine combinations in total. These values indicate relative strengths, such that ${\mathrm{prom}}_{H}\;=\;3$ represents a promoter nine times the strength of ${\mathrm{prom}}_{H}\;=\;\frac{1}{3}$. Furthermore, these values are chosen to align with the fold changes in strength that are typically found in part libraries [60,61]. For each combination, simulations are conducted with and without a slow codon, and the resulting steady-state $G_{rate}$ and $H_{rate}$ values are plotted in Figure 5.

It can immediately be seen that the general impact of a slow codon decreases both $G_{rate}$ and $H_{rate}$. The cause of this is rooted in ribosomal queue formation on ${\mathrm{mRNA}}_{H}$ heterologous transcripts, which we show by plotting the proportion of ribosomes on these transcripts that are on each footprint position (Figure 5d, ${\mathrm{prom}}_{H}\;=\;\frac{1}{3}$, ${\mathrm{RBS}}_{H}\;=\;3$). When using codons of uniform efficiency, ribosomes remain evenly distributed, while a slow codon at 26 $R_{f}$ produces a sharp rise in density upstream of this position, indicating queue formation. The slower translation that results from queue formation causes more ribosome sequestration on mRNA transcripts, reducing those available for translating other protein fractions. This wasteful ribosome sequestration on ${\mathrm{mRNA}}_{H}$ transcripts then leads to a reduction in both $H_{rate}$ and $G_{rate}$.

For both cases with and without a slow codon, it can be seen that higher synthetic gene expression from either increased ${\mathrm{prom}}_{H}$ or ${\mathrm{RBS}}_{H}$ leads to an increase in $H_{rate}$ and a decrease in $G_{rate}$. We plot this relationship in Figure 5c to further highlight the impact of ribosomal queue formation, which causes a more stringent inverse relationship between $G_{rate}$ and $H_{rate}$. Additionally, this relationship for the slow codon data is distinctly nonlinear, such that we see promoter-RBS combinations with equivalent values of $H_{rate}$, but different $G_{rate}$. We annotate three of these data points, highlighting how some combinations of ${\mathrm{prom}}_{H}$ and ${\mathrm{RBS}}_{H}$ are more efficient than others, i.e., they produce a higher value of $G_{rate}$ for the same value of $H_{rate}$.

#### 3.2.2. Identifying Optimal Gene Construct Designs by Quantifying Protein Production Yield Over Time

To provide a more thorough analysis of synthetic gene construct designs, we use $H_{rate}$ and $G_{rate}$ values from each promoter-RBS combination to calculate the heterologous protein yield over time (H(t)). In order to explore a range of construct design implications, we apply this to two cell growth scenarios: (i) uncapped exponential growth starting from a single cell and (ii) growth within a turbidostat at steady state where cell density remains constant. The protein yield H(t) is defined as the time integral of the product of $H_{rate}$(t) (the production rate per cell at time *t*) and N(t) (the number of cells at time *t*):(1)$$\begin{array}{c} H\left(t\right)\;=\;\int_{0}^{t}H_{rate}\left(T\right)N\left(T\right)\;\;dT. \end{array}$$

As such, H(t) represents the population-wide protein yield, rather than the protein yield per cell. The expression of N(t) can be changed to reflect the different growth scenarios that we propose. In both cases, we assume steady-state growth, so that the growth rate $G_{rate}$ and heterologous protein production rate per cell $H_{rate}$ remain constant over time, i.e., $G_{rate}\left(t\right)\;=\;G_{rate}\;=\;\mathrm{constant}$ and $H_{rate}\left(t\right)\;=\;H_{rate}\;=\;\mathrm{constant}$.

For uncapped exponential growth starting from a single cell, the number of cells at time *t* is given as $N\left(t\right)=2^{G_{rate}\;t}$. If we assume that there is no protein production at t=0, the protein yield at time *t* during steady-state exponential growth is given by:(2)$$\begin{array}{c} H{\left(t\right)}_{\exp}\;=\;\int_{0}^{t}H_{rate}2^{G_{rate}\;T}\;\;dT\;=\;\frac{H_{rate}}{G_{rate}\ln2}\left(2^{G_{rate}\;t}-1\right). \end{array}$$

For growth in a turbidostat, we assume that the cell population is already at steady-state density and that the turbidostat functions perfectly to keep cell density constant. Given this, the population size remains fixed over time such that N(t)=N=constant. If we again assume no protein production at t=0, the heterologous protein yield at time *t* within the turbidostat is given by:(3)$$\begin{array}{c} H{\left(t\right)}_{\mathrm{tur}}\;=\;\int_{0}^{t}H_{rate}N\;\;dT\;=\;H_{rate}N\;t. \end{array}$$

We furthermore note that $\frac{dH{\left(t\right)}_{\mathrm{tur}}}{dt}\;=\;H_{rate}N$, showing that the dynamics of heterologous protein yield in a turbidostat is time-invariant.

For each promoter-RBS combination, we calculate $H{\left(t\right)}_{\exp}$ and $H{\left(t\right)}_{\mathrm{tur}}$ and display their normalised values at successive time points in heat maps (Figure 6a,b), formally defined as $H{\left(t\right)}_{\mathrm{norm}}\left[i,j\right]={\left.\frac{H(t)[i,j]}{\max H(t)}\right|}_{{\mathrm{RBS}}_{H}=i,{\mathrm{prom}}_{H}=j}$. The promoter-RBS values considered in the heat maps of Figure 6b correspond to the promoter-RBS combinations considered in Figure 5a,b. Symmetry in heat map shading along the line ${\mathrm{prom}}_{H}\;=\;{\mathrm{RBS}}_{H}$ indicates that increasing either variable has the same effect on increasing protein yield, whereas a bias to one side indicates that one of the two variables (${\mathrm{prom}}_{H}$ or ${\mathrm{RBS}}_{H}$) has a greater effect. For the absolute values in both exponential and turbidostat cases, see Appendix A.

By comparing the heat maps over time, we can see how the gene construct’s design affects the value of $H{\left(t\right)}_{\mathrm{norm}}$. For values under the label “t=0 h”, we calculated protein yield over a small non-zero time interval ($10^{-12}\;h$) in order to observe how yields compare between construct designs at the start of production. For t=0 h, therefore, the population has just started to grow, and the burdensome impact of synthetic gene expression is minimal for both the “uniform efficiency” and “slow codon” cases. This means that maximising $H_{rate}$ through high ${\mathrm{prom}}_{H}$ and ${\mathrm{RBS}}_{H}$ would also maximise H(t) (Figure 6b, first column). As time extends, the population grows, and the negative impact of unnatural gene expression on the population’s growth becomes more extreme, such that achieving maximal $H{\left(t\right)}_{\exp}$ increasingly demands lower ${\mathrm{prom}}_{H}$ and ${\mathrm{RBS}}_{H}$ (Figure 6b, last column). In between these time points, different promoter-RBS combinations become optimal, as indicated by the changing location of the heat map quadrant with value “1”. For codons with uniform efficiency (top row, blue), this evolution from high to low expression is fairly symmetrical across promoter and RBS values, with only a slight bias towards the RBS, suggesting that increasing either parameter has an approximately equivalent effect. When ribosomal queues are present (bottom row, orange), however, there is a clear bias towards lower ${\mathrm{RBS}}_{H}$ and higher ${\mathrm{prom}}_{H}$, suggesting that maximising $H{\left(t\right)}_{\exp}$ over time requires the use of stronger promoters and weaker RBSs.

In the case of growth in a turbidostat, since the heterologous protein yield dynamics is time-invariant, the heat map values at t=0 h hold for all successive time points. This means that the negative impact on $G_{rate}$ from construct expression does not influence protein yield, suggesting that maximising construct expression (top-right quadrant) produces the optimal protein yield. As was seen in uncapped exponential growth at t=0 h, there is a strong bias towards a stronger promoter to maximise $H_{tur}$ when slow codons are present.

Beyond the qualitative trends observed over time in Figure 6b, we introduce a metric to quantify the effect that increasing the promoter strength or RBS strength has on increasing H(t). To this end, we implement a metric that considers all values of $H{\left(t\right)}_{\mathrm{norm}}$ for a given time point and outputs the extent to which they are weighted on either side of the line ${\mathrm{prom}}_{H}\;=\;{\mathrm{RBS}}_{H}$ in the corresponding heat map. Mathematically, by using our definition of $H{\left(t\right)}_{\mathrm{norm}}\left[i,j\right]$ introduced previously, we define $X\;=\;\sum_{j}\sum_{i}jH{\left(t\right)}_{\mathrm{norm}}\left[i,j\right]$, the sum of all $H{\left(t\right)}_{\mathrm{norm}}$ values weighted by ${\mathrm{prom}}_{H}$, and $Y\;=\;\sum_{i}\sum_{j}iH{\left(t\right)}_{\mathrm{norm}}\left[i,j\right]$, the sum of all $H{\left(t\right)}_{\mathrm{norm}}$ values weighted by ${\mathrm{RBS}}_{H}$. We then define the “Construct Score” as the difference between these terms, Y−X (see Figure 6a,c). Positive values of the Construct Score indicate a bias towards ${\mathrm{RBS}}_{H}$, meaning that increasing the RBS strength would result in a greater H(t), and hence a more efficient construct, than increasing the promoter strength by the same amount. The opposite holds true for negative values, where increasing the promoter strength would give rise to a greater H(t) than equivalently increasing the RBS strength. A value of zero, meanwhile, indicates perfect symmetry along the line ${\mathrm{prom}}_{H}\;=\;{\mathrm{RBS}}_{H}$, suggesting that there would be no discernible difference to H(t) if either ${\mathrm{prom}}_{H}$ or ${\mathrm{RBS}}_{H}$ were increased by the same amount. In this light, the Construct Score provides a useful metric to compare the effect of changing one part vs. another, but does not provide information on the quantity of protein yield itself.

Plotting the Construct Score for both codon cases over time reaffirms the trends seen from the heat maps. For uncapped exponential growth, in the case with no ribosomal queues, the Construct Score shows only a minimal bias towards the RBS, indicating that increasing ${\mathrm{prom}}_{H}$ or ${\mathrm{RBS}}_{H}$ would have similar effects on protein yield. In the case when a slower codon is introduced, there is strong bias towards the promoter. This suggests that one should increase ${\mathrm{prom}}_{H}$ to increase H(t), as increasing ${\mathrm{prom}}_{H}$ will yield a more efficient construct than increasing ${\mathrm{RBS}}_{H}$. Over time, the most efficient promoter-RBS combination increasingly becomes that which conveys the least burden on population growth, i.e., the bottom-left quadrant of the heat map (low ${\mathrm{prom}}_{H}$, low ${\mathrm{RBS}}_{H}$). This quadrant becomes more dominating over time relative to the other promoter-RBS combinations; therefore, the symmetry of $H{\left(t\right)}_{\mathrm{norm}}$ values around the line ${\mathrm{prom}}_{H}\;=\;{\mathrm{RBS}}_{H}$ becomes greater, and the Construct Score tends towards zero (Figure 6c, t=24 h). In the case of growth in a turbidostat, the heat map seen at t=0 h is maintained for all time points (i.e., negligible RBS bias without ribosomal queues and promoter bias with), and hence corresponds to horizontal lines in the time evolution of the Construct Score for both codon cases.

## 4. Discussion

We increased the efficiency of an existing TASEP framework by removing the possibility of selecting a queuing ribosome [56] and merged this modified TASEP with a stochastic implementation of the whole-cell model introduced in [4]. Using this modelling framework, we are able to simulate translation at the codon level while linking the effects of ribosomal queues to protein yield and cell growth. This leads to a number of implications for gene construct design, which are discussed below.

### 4.1. Implications for Gene Construct Design

We primarily explored how the sustained expression of a synthetic gene construct is coupled to cell growth through the re-distribution of finite cellular resources. In particular, we studied the relationship between promoter strength, RBS strength and codon efficiency in order to predict the optimal gene construct design for maximising protein yield. Our core results used slow codons that maximised the effects of ribosomal queues (slow codon with 0.5% efficiency located towards the end of a transcript of length 30 $R_{f}$). However, additionally, we also explored the effects of other codon features and report these in the Appendix A. In particular, we show how the relationship between $G_{rate}$ and $H_{rate}$ changes when considering slow codons with higher efficiency (3%), slow codons positioned towards the beginning of a transcript and longer mRNA transcripts (60 $R_{f}$).

While natural systems have been seen to sometimes use slow codons for positive growth effects (Section 1.2), we note that the use of slow codons in synthetic gene constructs would predominantly be burdensome to the host cell, either due to experimental constraints such as genetic stability or through unintentional placement. We therefore began our analysis by showing how slow codons negatively impact cell growth and heterologous gene expression through ribosomal queue formation. This highlights the general importance of optimising codon efficiencies. Achieving this is often difficult due to the varied effects of gene expression burden and context-dependent expression [62]. In light of this, we explored how other aspects of gene construct design can be optimised when faced with a codon composition that triggers significant ribosomal queuing.

Different promoter-RBS combinations were seen to yield higher growth rates for equivalent values of $H_{rate}$, suggesting that the optimum design choice can change when ribosomal queues exist. To explore this further, we devised a metric to compare whether increasing promoter strength (${\mathrm{prom}}_{H}$) or RBS strength (${\mathrm{RBS}}_{H}$) by the same amount had equivalent or different effect on increasing the protein yield. We then applied this to uncapped exponential and turbidostat growth at steady state. Without ribosomal queues, we found that increasing ${\mathrm{RBS}}_{H}$ has a minimal added benefit on the heterologous protein yield over increasing ${\mathrm{prom}}_{H}$. This could be a result of increased “ribosome protection”, which prevents the degradation of ribosome-bound mRNAs, as a lack of queuing ribosomes on one transcript would increase the chance that all transcripts have at least one protective ribosome. This would therefore boost the overall translation capacity for heterologous proteins. When queue formation occurs, however, increasing ${\mathrm{prom}}_{H}$ was seen to be significantly more beneficial for heterologous protein yield than increasing ${\mathrm{RBS}}_{H}$. Such scenarios could occur due to an imbalance between free ribosomes and mRNA transcripts in the cell. In these cases, increasing ${\mathrm{prom}}_{H}$ would increase the amount of mRNAs that free ribosomes can translate, thus distributing the load and reducing potential queues. A higher ${\mathrm{RBS}}_{H}$, meanwhile, would force more ribosomes onto existing transcripts and thus heighten queue formation. Above all, this analysis suggests that the ability to control transcription or translation independently of each other, and hence control the allocation of different resource pools, would be an extremely valuable experimental tool. This is an approach that is increasingly being considered in synthetic biology designs, as illustrated by [63].

The “promoter over RBS” design principle that we identify is one that has seen experimental support [1]. Furthermore, the notion that the least burdensome designs convey maximal protein yield in the long-term (due to an enhanced population growth rate) has also been observed experimentally and has subsequently been used to motivate the development of tools to regulate burden within a cell [64]. Our results echo this, showing that a switch from more- to less-burdensome designs over the time course of an experiment would maximise protein yield. This analysis could furthermore be used as a basis to predict the experimental time range over which a particular gene construct design could deliver optimal protein expression, although accurately achieving this would require finer modelling and additional experimental evidence.

For growth in a turbidostat at steady state, we previously noted that the dynamics of H(t) are time-invariant, suggesting that any implications for construct design on protein yield can be seen from the results at t=0 h. For both codon cases, our results suggest that optimal protein yield can be obtained by maximising both promoter and RBS strengths, as shown by the top-right quadrant of the first column of the heat maps in Figure 6b. When ribosomal queues are present (bottom row, orange), our results suggest that increasing promoter strength would generally have a stronger effect on boosting protein yield compared with an equivalent increase of RBS strength. Despite this analysis, we note that strong construct expression may also promote other negative consequences of cellular burden, such as mutation accumulation due to genetic instability, which StoCellAtor does not consider.

### 4.2. Future Applications of StoCellAtor

A natural way to expand the remit of StoCellAtor’s results would be to consider the effects of more complex codon distributions along an mRNA transcript, and in doing so, explore the notion that slow codons can be used for positive growth effects. In Section 1.2, we noted how organisms have been seen to use 5’ “ramp up” zones that decrease the likelihood of costly upstream ribosome collisions and wasteful ribosomal queues [13,14,15] or slow regions that increase the fidelity of cotranslational folding [16,17]. Such features may be equally desirable in synthetic gene constructs, and so, a natural extension of StoCellAtor would be in predicting the most efficient “ramp up” designs or “slow regions” when using different combinations of promoters and RBSs. We note that existing codon-optimisation tools are able to simulate complex codon designs, most notably the biophysical model of [21]; however, these are all disconnected from a WCM setting with a resource-dependent account of the growth rate. We demonstrate a simple version of the ramping effect by positioning a single slow codon towards the 5’ end of the synthetic transcript (Figure S1). We note, however, that these preliminary simulations require further exploration.

A broader future application would involve addressing a previously referenced shortcoming of our model’s predictions and requires looking at the role of burden and construct design on genetic instability. In typical experimental settings, when expressing synthetic gene constructs over time, they inevitably accumulate mutations, causing decreased expression and/or complete construct failure. Predicting the dynamics of mutation spread and its impact on protein expression is a complex problem, for which gene expression burden [65] and DNA sequence composition [29] are known to play major roles. However, such analyses fall short of accurately predicting mutation spread dynamics, because they do not consider them within a “whole-cell” context. For a given protein expression system, being able to quantify burden and link its effect to growth rate is therefore important in informing how mutations propagate.

In order to address this problem, and thereby link StoCellAtor to a description of mutation dynamics, one suggestion we are currently exploring is to first subdivide the bacterial cell population used in our model into two sub-populations: an “engineered” variety that grows more slowly and a “mutant” that has lost capacity for construct expression due to a fatal mutation, for example within its promoter or RBS region. An engineered cell would be able to mutate into a mutant with a particular transition probability, and each cell type would have an associated growth rate calculated from our model. This could then be used to inform how quickly one sub-population is selected for comparison with the other. As the mutant cells cannot express their construct, they would carry less burden than the engineered cells and thus grow faster. As seen from our results, the design of the gene constructs in the engineered cell would strongly influence burden, and this would hence dictate how fast one sub-population grows relative to another. In the case of turbidostat growth, where cell density is kept constant, this would lead to a complete out-competition of engineered cells over time, something that has been well-documented experimentally [66]. These considerations, which depend on having a strong grasp on the cellular processes that contribute to burden, would therefore be vital to be able to predict protein yields in continuous cultures.

Regardless of the specific use-cases presented here, we hope that the modelling framework we have introduced here will encourage its users to consider the impact of construct design on cellular resources and population dynamics and, through this, allow them to computationally explore designs that minimally impact growth and optimise synthetic expression yields.

## Figures and Tables

**Figure 1 biology-10-00037-f001:**
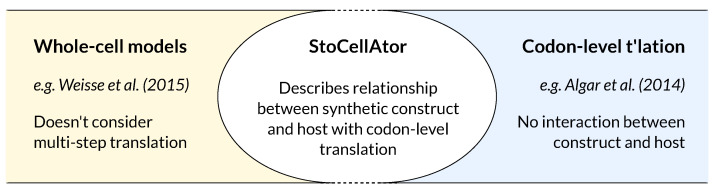
Our model combines codon-level translation with a resource-based whole-cell model, two components that have previously not been joined. In doing so, we see how the effects of ribosomal queues impact the relationship between synthetic construct expression and the dynamics of cell growth and heterologous protein expression and yields. StoCellAtor, stochastic whole-cell calculator.

**Figure 2 biology-10-00037-f002:**
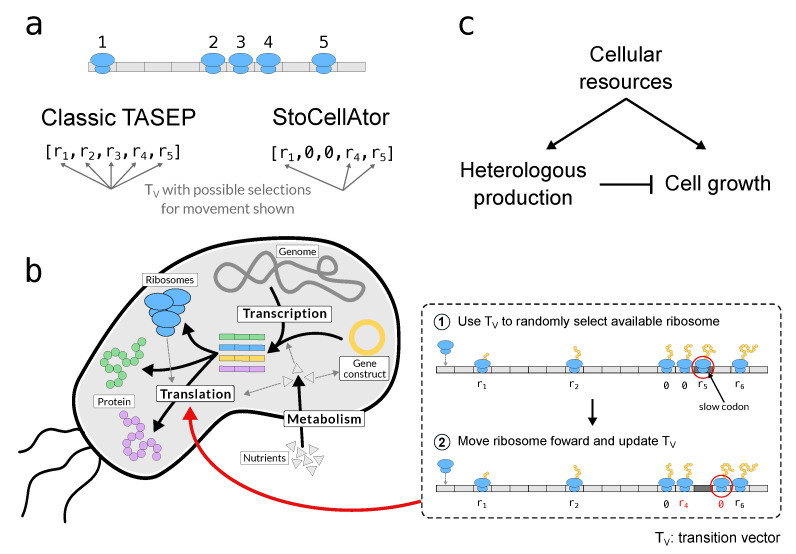
StoCellAtor’s translation model in context. (**a**) The difference between classic TASEP and StoCellAtor in terms of choosing ribosome movement via the transition vector (T_V_). (**b**) The simulation steps taken during translation in the context of a resource-limited whole-cell model, which considers nutrient metabolism, transcription and translation. Step 1: a non-queuing ribosome is selected for movement. Step 2: the chosen ribosome position is updated. This ribosome might become “queuing”, while the ribosome behind it becomes free to move. This is reflected in the updated T_V_ (red values). (**c**) A top-level summary of the whole-cell model, showing the links among the cell’s resources, its heterologous protein production and its growth. The activation and inhibition arrows denote general effects and not specific reactions.

**Figure 3 biology-10-00037-f003:**
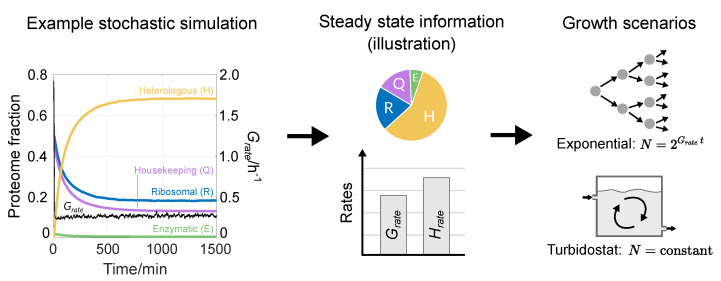
How we apply StoCellAtor to relevant growth scenarios. (**Left**) An example stochastic simulation of the different proteome fractions (left y-axis) and growth rate (right y-axis) with ${\mathrm{prom}}_{H}\;=\;3$ and ${\mathrm{RBS}}_{H}\;=\;1$. Values start out of equilibrium, go through transient dynamics and finally reach steady-state values. (**Middle**) An illustration of the steady-state information gained from each simulation. (**Right**) Steady-state information is used to assess protein production in a hypothetical population that grows over time. Two growth scenarios are considered: uncapped exponential growth and growth within a turbidostat.

**Figure 4 biology-10-00037-f004:**
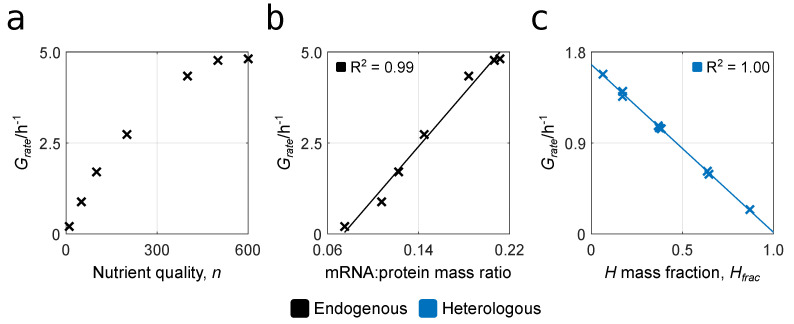
Characterising StoCellAtor’s behaviour in both endogenous and heterologous simulations. (**a**) Recovering Monod’s law: the hyperbolic dependency between external nutrient quality and growth rate. (**b**) Recovering Schaechter’s law: the linear relationship between growth rate and the mRNA:protein mass ratio. A different value of nutrient quality (corresponding to the values in subfigure a) is used for each data point. (**c**) Recovering the linear relationship between $G_{rate}$ and $H_{frac}$ that was experimentally observed in [3]. For each data point, different combinations of promoter and RBS strengths are considered (see Section 3.2), while the nutrient quality parameter is fixed to n=100. A linear regression with corresponding $R^{2}$ values is also shown.

**Figure 5 biology-10-00037-f005:**
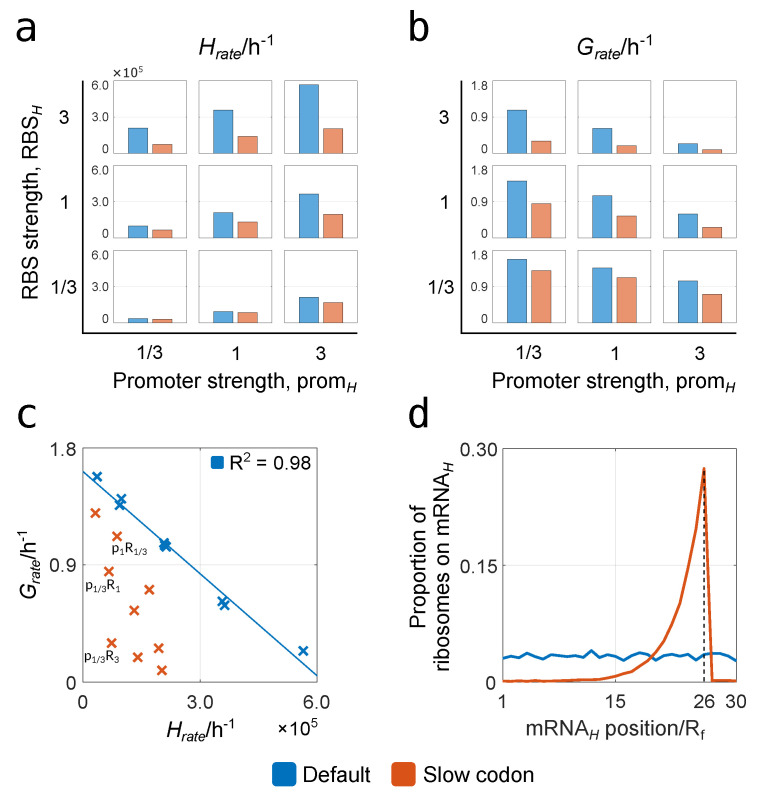
How gene construct design and inefficient codons affect performance. Blue represents the case when all codons on the gene construct have the same efficiency, while orange represents the case when a codon with lower relative efficiency (efficiency of 0.5% compared to the other codons) is introduced at position 26 $R_{f}$. All simulation results used a fixed nutrient quality of n=100. (**a**) The effect of heterologous promoter and RBS strength on $H_{rate}$. (**b**) The effect of heterologous promoter and RBS strength on $G_{rate}$. (**c**) The relationship between $G_{rate}$ and $H_{rate}$. Three results with similar $H_{rate}$ values are highlighted with relative values of ${\mathrm{prom}}_{H}$ (p) and ${\mathrm{RBS}}_{H}$ (R) indicated. (**d**) Proportion of ribosomes on ${\mathrm{mRNA}}_{H}$ that are on each footprint position for a gene construct with low ${\mathrm{prom}}_{H}$ (${\mathrm{prom}}_{H}=\frac{1}{3}$) and high ${\mathrm{RBS}}_{H}$ (${\mathrm{RBS}}_{H}=3$).

**Figure 6 biology-10-00037-f006:**
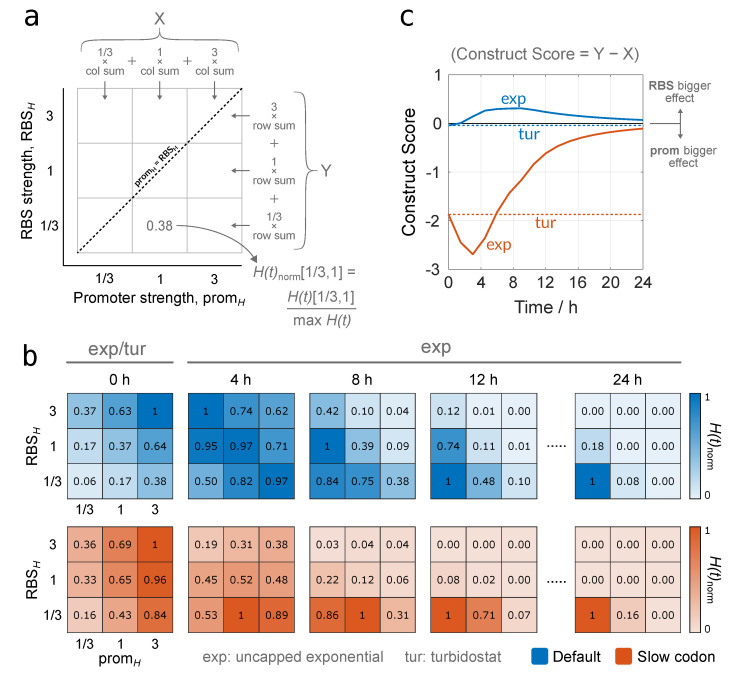
Evaluating the performance of gene construct design in terms of heterologous protein yield by calculating $H{\left(t\right)}_{\mathrm{norm}}$ in different growth scenarios. (**a**) An illustration of the grid space used to form heat maps of $H{\left(t\right)}_{\mathrm{norm}}$ (subfigure b) and the Construct Score (Subfigure c). The line ${\mathrm{prom}}_{H}\;=\;{\mathrm{RBS}}_{H}$ is used to compare the effects of ${\mathrm{prom}}_{H}$ and ${\mathrm{RBS}}_{H}$. (**b**) Heat maps of $H{\left(t\right)}_{\mathrm{norm}}$ at successive time points for cases without (blue) and with (orange) a slow codon. For the label “t=0 h”, calculations are made over a small non-zero time interval ($10^{-12}\;h$). Beyond t=0 h, we provide heat maps only for $H{\left(t\right)}_{\exp}$ because the dynamics for $H{\left(t\right)}_{\mathrm{tur}}$ remain unchanged. (**c**) Comparing the effect of promoter and RBS strengths on $H{\left(t\right)}_{\exp}$ and $H{\left(t\right)}_{\mathrm{tur}}$ via the “Construct Score”. A value of zero indicates that increasing ${\mathrm{prom}}_{H}$ and ${\mathrm{RBS}}_{H}$ by the same amount would have an equivalent effect on protein yield, while positive and negative values indicate that a greater effect on yield would be obtained by increasing ${\mathrm{prom}}_{H}$ or ${\mathrm{RBS}}_{H}$, respectively. Dashed lines indicate the Construct Score when using a turbidostat.

## Data Availability

All data obtained from our simulations is available at https://doi.org/10.5281/zenodo.4415761.

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
