# Peer review of "A Modelling Framework Linking Resource-Based Stochastic Translation to the Optimal Design of Synthetic Constructs"

_biology, 2021, doi:10.3390/biology10010037_

Round 1

Reviewer 1 Report

In this manuscript the authors propose an extension of existing burden-aware models of heterologous gene expression to also include consideration of codon-level translation dynamics.  They analyse the interplay between Promoter and RBS strength in the presence/absence of a “slow” codon, and highlight how these factors combine to optimise both growth and heterologous protein yield. I found the manuscript interesting, timely, and it represents an insightful extension of the Authors’ past theoretical and experimental work. Overall I thought the paper was high-quality and rigorous, and so would like to recommend it for publication following the authors’ consideration of the following (mostly minor) points.

General points:

  1. The authors state that they focus their analysis on a “scenario that heightened the effects of ribosomal queues” (Line 91) by using a slow codon with efficiency 0.5%. I think this choice (of a very low efficiency codon) is fair to highlight behaviours they are investigating. But, I believe the paper might benefit from also providing some analysis of cases where the slow codon efficiency isn’t quite so small (Say, 5%). In such a case would the system behave pretty much like the no-slow-codon case? Or is there still a significant difference (e.g. in Fig 5d and 6c) due to the ribosomal interactions? If it is too much work to do all the analysis again for such a case, perhaps the authors could at least describe the anticipated behaviour in the Discussion section.
  2. The “Impact Score” metric could be more clearly defined. There is some discussion of this in Line 188, and at the end of Figure 6’s caption – but I didn’t find either explanation particularly intuitive. It is more rigorously defined Line 208, but as the score is already being discussed this definition comes a bit too late. Essentially, at first reading I was confused as to whether a high Impact Score was a good thing (i.e. it increases H(t)) or a bad thing (i.e. it places more burden on the cell). My understanding was not helped by the fact that throughout the manuscript the word “impact” is used both in the name of this metric, and also interchangeably to discuss reductions in cell growth rate (i.e. Line 73), influences of codons on ribosomal queues (Line 4), and burden or protein yield (Line 44). Some of these are “good” things, and some “bad” (from the point of our would-be genetic engineer) – so when I then had to consider “Impact Scores” I did not know whether I wanted them to be high or low! It would be ideal if the definition of this score came earlier (i.e. before Line 208), and perhaps consider devising a different name for the “Impact Score” (to avoiding ambiguous use of the word “Impact”).
  3. Again regarding the impact score – I found the example starting line 211 somewhat unclear. Here it says (for example) that a positive impact score means RBSh gives a greater contribution to increasing H(t). Does this mean that the impact score at that point in RBS/Promoter parameter space has a greater dependence on RBS strength? (i.e. the statement is saying that we have d[Impact Score]/d[RBSh]> d[Impact Score]/d[promoter] at that point?).
  4. If slow codons cause so many issues, why do they exist? I have seen several theories proposed in the past, for example that by causing ribosomes to temporarily stall they give time for sections of the partially translated protein to fold. It would be good if the authors could cite some more relevant literature (perhaps around Line 239 where the negative impact of slow codons is discussed). After all, maybe there are cases in which a genetic engineer might want temporary stalling in translation.

Other minor points:

  1. Line 13 – The phrase “while also seeing how these implications change when faced with different codon efficiencies.” Could be made more precise, i.e. to specify that you investigate the presence of a low efficiency codons, rather the impact of codon efficiency in general (which you don’t).
  2. Fig 5C – Given how few data points there are here, and the fact that the Slow Codon case looks significantly non-linear, I am not convinced that fitting linear trend-lines is particularly meaningful. You even state this yourself around line 169 (that there is interdependence between parameters for the slow codon case – hence why they don’t form a linear relationship). Maybe the trend lines could be removed and the differences in linearity just highlighted in the text of 3.2.1? Alternatively, would it be possible to run many more simulations with intermediate prom_H and RBS_H values to better populate this plot, so the underlying trends are clearer?
  3. Would you kindly add labels to the colour scales on the right of Fig6b.
  4. Line 248 – This discussion of RBS strength importance could be tightened up. Looking at Fig5a I would have concluded that without a slow codon, increasing RBS strength does have significant benefit on heterologous protein yield.
  5. Line 286 – Authors mention using StoCellAtor to analyse and predict codon “ramp up” designs for synthetic constructs – do any existing codon optimisation tools do this?

Reviewer 2 Report

The manuscript describes an expansion to whole cell models to account for ribosomal jamming by incorporating the frequently used TASEP model for ribosomal queues into an existing whole cell model. In order to do this, the authors converted the deterministic whole cell model to a stochastic model. They also introduced a speed up to their implementation of the traditional TASEP model referenced, by only considering ribosomes that are able to progress to speed up simulation efficiency. They then utilize the model to examine the case of a synthetic gene with and without a slow codon, a potential source of ribosomal jamming. By looking at different promoter and RBS strengths, they suggest optimal combinations of promoter and RBS strengths to decrease the cellular burden due to ribosomal jamming. The paper is generally well written, though some of the description in the results is a bit confusing (see comments). As synthetic circuits increase in complexity, their burden on cells increases. As such, modelling frameworks to determine optimal circuit design to reduce overall cellular burden are welcome such as those described in the paper are welcome. However, I believe that more work has to be done in order for this manuscript to be suitable for publication. Specific things I have in mind are:

1. The introduction can be expanded to include more of a context for biophysical models that include ribosomal jamming or can be expanded to this (see review by Zur and Tuller in NAR)

2. A better demonstration that the model is able to recapitulate experimental results aside from Monod's and Schaechter's Laws that highlights the advantage of this model over just whole cell models or biophysical models that account for ribosomal stalling.

3. I found section 3.2.2 confusing. Is H(t) seems to be the total protein yield at time t, this supported by equation (1), (2), and (3) shouldn't this be strictly increasing as time increases? Why are there zeros in figure 6b? The turbidostat yield at time=0 should also be zero if H(t)_tur = H_rate N t. This section needs to be described more carefully.

4. Can some of the whole cell stochastic models be adjusted to include ribosomal queuing?

Reviewer 3 Report

This manuscript describes the modeling framework for the cell growth based on the resource-based stochastic translation with RBS strength and the codon efficiency for the optimal design of synthetic constructs.

  1. Although the computational pat can be understood, it is not clear about how the metabolism (including metabolic regulation) was incorporated in the whole-cell model (genome-scale model is not shown in the supplemental file). I wonder how the growth low could be properly modeled ?
  2. Quality of the nutrients depend on the carbon sources and other nutrient sources including nitrogen, phosphate, sulfur , ion sources used, that affect the cell growth rate. How is the ‘nutrient quality’ defined in the present article ?
  3. As the nutrient quality increases, the cell growth rate increases (as implied in Fig.4a), as well as the ribosomal fraction, while enzymatic/catabolic protein fraction must decrease, while the housekeeping protein fraction may be constant. Are these characteristics confirmed by the simulation ? What is exactly mean ‘heterologous protein’ in the present MS ?
  4. Although the changes in the proteome factions is given with respect to time in Fig.3a, is this result confirmed by the experimental data ? Please show the steady state values with respect to the cell growth rate.
  5. The advantage of whole cell modeling approach is to compute the amino acid synthesis for the protein synthesis and the cell growth rate by properly modeling the translational machinery. I wonder if the present model could attain codon optimization for maximizing the cell growth rate ?
  6. In relation to the translational activity in response to the nutrient quality, RelA-SpoT control the (p)ppGpp level and regulate the translational activity, but it is not mentioned at all about this in the present MS.
  7. In the present simulation, both uncapped exponential growth and turbitostat were considered. I wonder why the standard chemostat was not considered instead ?In conclusion, the above comments may be clarified before acceptance.
  8.  

Round 2

Reviewer 2 Report

I would like to thank the authors for their revised manuscript in response to my comments. They have mostly been addressed, I have a few points remaining for the authors to consider:

  1. The clarification on H_norm was very helpful and it is a lot less confusing now. However, I think it is still worth reporting the absolute H_tot values across all times. The reason for this is if this tool is used for synthetic circuit design a design goal may be the maximum amount of H without regard to final cell fitness. In this case, one can imagine a situation where maximizing the promoter and/or RBS strength would be optimal even with the growth penalty, than a circuit with weaker transcription/translation that is able to grow for longer. Is there a way to use the model to show this?
  2. Relatedly, is there a way to use the model to predict an optimal promoter/RBS strength combination that maximizes H_total?
  3. A related paper came out recently that uses a mean field approximation of TASEP, which allows the use of differential equations (as opposed to stochastic simulation), while not a whole cell model, they used the amount of free ribosome as a proxy for cell fitness and it seems to have predictive power with experimental verification, while not explicitly using synthetic circuits, it would be good to reference this paper and discuss how this fits in the context of this research (https://www.nature.com/articles/s41598-020-78260-y) 

Reviewer 3 Report

The manuscript has been appropriately modified, and this may be now accepted.

Author Response

We warmly thank the reviewer for his constructive and very helpful comments and are glad that we have been able to address all the points that were previously raised.